# Enhanced Semantic Alignment in Transformer Tracking via Position Learning and Force-Directed Attention

## Abstract

In the field of visual object tracking, one-stream pipelines have become the mainstream framework due to its efficient integration of feature extraction and relationship modeling. However, existing methods still face the issue of semantic misalignment: firstly, the interaction of position encoding between the two branches leads to a misalignment between feature semantics and position encoding; secondly, traditional attention mechanisms fail to distinguish between semantic attraction and repulsion among features, resulting in semantic misalignment when the model processes these features. To address these issues, we propose an Enhanced Semantic Alignment Transformer Tracker (ESAT) with position learning and force-directed attention. By leveraging self-supervised position loss (SPL), ESAT separately learns the absolute position encodings of the target and search branches, distinguishing the locations of various tokens and their positive or negative relationships, thereby enhancing the semantic consistency between position and features. Additionally, ESAT incorporates a repulsion-attraction mechanism applied to the self-attention module, named force-directed attention (FDA),simulating dynamic interactions between nodes to improve feature discrimination. Extensive experiments on multiple public tracking datasets show that our method outperforms many pipelines and achieves superior performance on five challenging benchmarks.

## 1 Introduction

Visual Object Tracking (VOT) has attracted increasing attention due to its broad applications in various fields, including traffic monitoring (Chandrakar et al., 2022), medical science (Bouget et al., 2017) and self-driving cars (Gao et al., 2019). Single Object Tracking (SOT) is a one of the popular category of VOT, which can be described as follows: Capture the target's appearance features in the first frame and maintain continuous tracking in subsequent frames (Soleimanitaleb & Keyvanrad, 2022). Notwithstanding many highly effective trackers have emerged (Bertinetto et al., 2016; Li et al., 2019), challenges such as occlusions and deformations continue to perplex numerous researchers, impinging on the accuracy and stability of tracking. In recent years, based on Transformer (Vaswani et al., 2017), Vision Transformer (ViT) (Dosovitskiy et al., 2020) with its superior attention mechanisms and capacity for global feature capture, have achieved remarkable success in VOT tasks (Chen et al., 2022; Lin et al., 2022; Hu et al., 2024).

Despite advancements in feature fusion, current prevailing trackers still have limitations in dealing with semantic alignment. **On the one hand**, self-attention cannot capture the ordering of input tokens, so incorporating positional encoding is particularly important. In VOT tasks, absolute positional encoding (Vaswani et al., 2017) is a widely adopted method. This method assigns distinct vectors to each position in the template and search token sequences, which may result in a semantic misalignment. Meanwhile, the template contains target feature information, while the search region includes potential target positions and background information. However, the differing semantic content and context of these areas may hinder the ability of model to effectively capture their relationships. **On the other hand**, traditional attention mechanisms in object tracking exhibit a significant limitation in feature weight allocation. Traditional methods indiscriminately sum the weights of various features, overlooking the complex interactions between them. This "one-size-fits-

all" approach is particularly inadequate given the semantic complexity, where feature relationships involving appearance, motion patterns, and background are interwoven and interdependent. For instance, in SOT tasks, certain features like the target's color may attract each other in specific contexts, while others, such as background interference, may repel due to semantic conflicts. Traditional attention mechanisms struggle to capture these subtle differences and dynamic relationships, leading to biases in the model's tracking accuracy and ultimately affecting its ability to discern the target's true state and motion trajectory.

To alleviate these problems, we propose a novel **Enhanced Semantic Alignment Transformer Tracker (ESAT)** with position learning and force-directed attention. Specifically, we introduce a novel self-supervised position loss function and a self attention mechanism incorporating repulsive and attractive forces. **(1)** Self-supervised position loss (SPL) allows each token to implicitly incorporate its positional information. We propose two types of positional labels: absolute positional labels and positive-negative positional labels. We input the features of the template and search regions outputted by the tracking model into two separate fully convolutional networks (FCN), each consisting of $L$ stacked Conv-BN-ReLU layers. This module outputs the corresponding position information for each image patch. Based on this, we calculate the position losses using the two types of labels and the obtained positional information separately, distinguishing the positions of different tokens and their positive and negative values, to enhance the semantic consistency of positions and features. **(2)** Force-directed attention (FDA) define the repulsive and attractive forces, by calculating the difference between query (q) and key (k). Repulsive force reduces the attention weight between different features, while attractive force enhances the weight of similar features. By adjusting the attention matrix, the model considers the interaction between features in attention allocation, thereby reducing the problem of semantic misalignment.

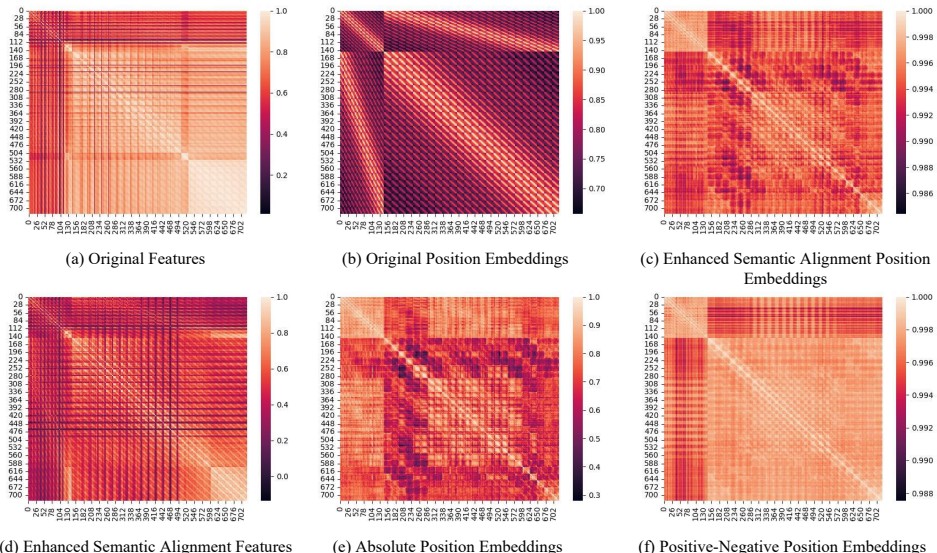

Figure 1: Visualization of the correlation between features of the template and search region, as well as their positional correlation. The lighter the color indicates the higher the correlation; conversely, the darker the color indicates the lower correlation. The visualization of correlations can be divided into four parts: template region self-correlation, search region self-correlation, and two sections representing the correlation between the template and the search image.

As depicted in Fig.1 (a), the original features have a smaller color distinction, while the features of the ESAT exhibit a greater color distinction, which indicates that the semantics of the model have been enhanced after applying ESAT shown in Fig.1 (d). Both template and search images start from 1, which can cause the most relevant areas to shift towards the beginning when the two regions interact, resulting in a misalignment of semantics, as illustrated in Fig.1 (b). Therefore, we apply self-supervised position learning to make the template most relevant to the target position in the search area. The learned absolute positional embeddings indicate that the most relevant regions for both are located in the middle, which corresponds to the areas where the template and the target po-

sition in the search region are most related. The positive-negative positional embeddings correspond to the absolute, highlighting the most relevant areas. By merging these two, we obtain the enhanced semantics alignment position embeddings as seen in Fig.1 (c).

To sum up, the main contributions of this work are threefold: **(1)** We propose a self-supervised position loss (SPL) and two positional labels: absolute and positive-negative positional labels, which mitigates the issue of semantic misalignment between template and search region. **(2)** We introduce force-directed attention (FDA) to adjust the attention weight matrix. FDA can enhance attention between similar features and weaken attention between different features to improve the semantics between features. **(3)** The above modules collaboratively form a comprehensive Enhanced Semantic Alignment Transformer Tracker (ESAT). Experiments on several widely used benchmarks demonstrate its outstanding performance and validate the efficiency and effectiveness of SPL and FDA.

## 2 RELATED WORK

### 2.1 TRANSFORMER BASED TRACKER

Transformer-based tracking models can be categorized into two distinct groups based on the framework and method of feature extraction they employ: CNN-Transformer based trackers and Fully-Transformer based trackers (Kugarajeevan et al., 2023).

**CNN-Transformer based trackers** (Chen et al., 2021b; Yan et al., 2021b) combine the Convolutional Neural Network (CNN) (Krizhevsky et al., 2012) and Transformer (Vaswani et al., 2017) to be a hybrid architecture. Following fully-convolutional siamese networks based trackers (Bertinetto et al., 2016; Li et al., 2019), they use two identical pipelines of CNNs to extracted the features of the target template and search region and then flat the features into vectors and pass this information to the Transformer which capture the similarity features of the target in the search region. Although CNN-Transformer based trackers utilize the attention mechanism, they still rely on CNN for feature extraction, which makes them to be difficult to capture global feature representations.

**Fully-Transformer based trackers** (Lin et al., 2022; Cui et al., 2022; Ye et al., 2022; Hu et al., 2024) are proposed to solve this problem, which are categorized into Two-stream Two-stage trackers and One-stream One-stage trackers. Two-stream Two-stage trackers use two identical and independent Transformer based tracking pipelines to extract features and then employ another Transformer network to find the relationships between these features. STMTrack (Fu et al., 2021) introduces a space-time memory network that integrates historical template and search features for better adapting to appearance variations during tracking. One-stream One-stage trackers like SimTrack (Chen et al., 2022) and OSTrack (Ye et al., 2022), using pretrained ViT as the backbone Transformer. OS-Track points out that some search patches contain background information, which is not necessary in the tracking process, so an early candidate elimination strategy gradually discards the background regions in the search area during the inference process. ARTrack (Wei et al., 2023) is proposed to treat tracking tasks as a coordinate sequence interpretation task, gradually estimating object trajectories. AQAT (Xie et al., 2024) uses a simple autoregressive query to effectively learn spatial-temporal information, capturing the instantaneous appearance changes of the target in a sliding window manner.

Although the dual stream method has achieved great success, the separation of feature extraction and relationship modeling has the following limitations. In this case, we use the One-stream One-stage method as as the baseline model.

### 2.2 ATTENTION MECHANISM IN TRANSFORMER TRACKING

The attention mechanism can establish rich global contextual dependencies, excelling at extracting edge features and distinguishing similar features, making it highly suitable for single object tracking tasks.

**CNN-Transformer based trackers:** STARK (Yan et al., 2021b) captures the feature dependency relationships between each element in the tracking sequence and enhances the original features using global contextual information, using multiple self attention layers to achieve information exchange and fusion between features. TransT (Chen et al., 2021b) replaces the traditional cross correlation

with attention, using self attention and cross attention based feature enhancement module in the feature fusion stage.

**Fully-Transformer based trackers:** For Two-stream Two-stage methods, building on TransT (Chen et al., 2021b), SparseTT (Fu et al., 2022) proposes a sparse multi-head attention mechanism to improve the ability to differentiate between foreground and background. SwinTrack (Lin et al., 2022) explore to concatenate template and search features, and use multiple self-attention layers to facilitate feature interaction and fusion, reducing computational costs and enhancing model accuracy. CSWinTT (Song et al., 2022) proposed an attention module based on multi-scale cyclic moving windows, elevating pixel level attention to window level, which helps to maintain the integrity of the target and preserve more bit information. Force reduces the ambiguity of the target edge area. For One-stream One-stage methods, MixFormer (Cui et al., 2022) designs a set of Mixed Attention Modules based on the CVT (Wu et al., 2021a) structure to simultaneously extract and integrate features of the target and search region. Compared to CNN-Transformer based trackers, Fully-Transformer based method has a larger network capacity, stronger representation ability for its output features, and it exhibits greater robustness.

### 2.3 POSITIONAL ENCODING

Transformer (Vaswani et al., 2017) relies on positional encoding (PE) to help it understand the spatial structure of images, as it does not utilize convolutional operations to extract local features. PE provides explicit spatial location information for each patch in the image. Each offering different advantages depending on the context of the task.

**Absolute Position Encoding** (APE) assigns a unique position vector to each element in the sequence or image, indicating its specific location. It provides a clear reference for the position of each element. There are several choices of APE. In Vision Transformer (ViT) (Dosovitskiy et al., 2020), the PE is generated by the fixed 1-D sinusoidal functions of different frequencies. At the same time, there are 2-D sinusoidal PE methods (Raisi et al., 2021; Wang & Liu, 2021) that incorporate information about the height and width of the image. Meanwhile, the APE can also be learnable, as it is randomly initialized and updated with the model's parameters during the training process.

**Relative Position Encoding** (RPE) encodes the relative distances or relationships between elements instead of focusing on the absolute position. It allows the model to understand how far apart elements are from each other. Based on the APE, RPE is added into the self-attention weight calculation. (Bello et al., 2019) firstly proposes a new relative positional encoding methods dedicated to 2D images, which is used in Swin-Transformer (Liu et al., 2021). (Wu et al., 2021b) further improves the 2-D RPE, called image RPE (iRPE), which integrates the modeling of directional relative distances as well as the interactions between queries and relative positional embeddings in self attention mechanism.

We build upon the previous method of explicitly combining positional information by using positional information as a supervised signal for the self-supervised training of the tracking model, allowing each encoded patch to implicitly contain its positional information.

## 3 METHODOLOGY

In this section, we introduce the framework of the proposed Enhanced Semantic Alignment Transformer Tracker (ESAT). Subsequently, we present self-supervised position loss and force-directed attention one by one.

### 3.1 FRAMEWORK OF ESAT

One-stream One-stage trackers integrate features more efficiently within a single Transformer, allowing simultaneous extraction and fusion. Therefore, we conduct research based on OSTrack (Ye et al., 2022). The pipeline of the tracker ESAT is shown in Fig.2. The whole framework can be divided into three parts:

**Positional Label Generation.** As shown in Fig.2 (a), we generate the corresponding absolute positional labels for each patch in the search image, while the absolute positional labels for the template

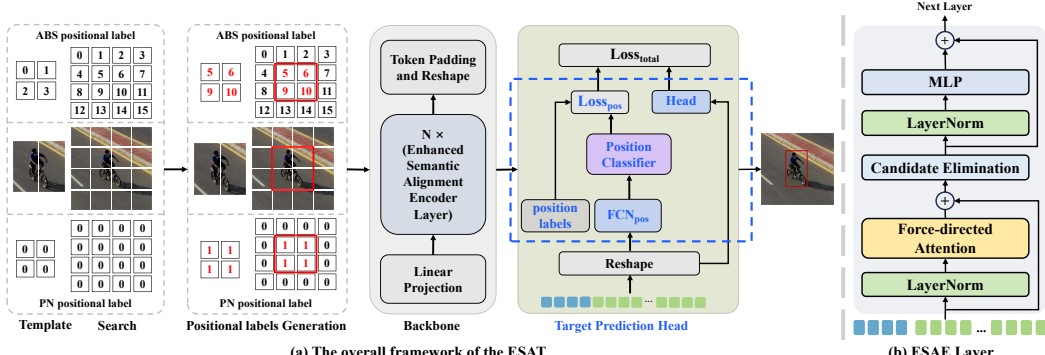

Figure 2: The pipeline of EAST.

image correspond to the central region of the search region. Next, we set the area corresponding to the ground truth box in the search image as the positional label of the template. For the positive-negative positional labels, we assign a value of 1 to both the template region and the target area in the search region, while the background in the search region is set to 0.

**Backbone.** The input of the ESAT pipeline are template image $z$ and search image $x$, and then split and flatten them into a series of patches $z_p \in \mathbb{R}^{N_z \times (3 \cdot P^2)}$ and $x_p \in \mathbb{R}^{N_x \times (3 \cdot P^2)}$, where $N_z$ and $N_x$ are respectively the number of template and search patches and $P \times P$ is the resolution of each image patch. Then, a Linear Projection Layer is used for generating the embeddings $E_z \in \mathbb{R}^{N_z \times D}$ of $z_p$ and $E_x \in \mathbb{R}^{N_x \times D}$ of $x_p$, where D is the embedding dimension. Next, we input them into several subsequent Encoder Layers. Learnable 1D position embeddings $P_z$ and $P_x$ are added to the patch embeddings of the template and search region separately to produce the final template token embeddings $H_z \in \mathbb{R}^{N_z \times D}$ and search region token embeddings $H_x \in \mathbb{R}^{N_x \times D}$. After that, all these tokens are concatenated as a sequence with a length of $N_z + N_x$ and fed to an encoder. Each encoder layer updates the input tokens via a Force-directed Attention (FDA) block and a feed-forward network (FFN) as illustrated in Fig.2 (b).

**Head and loss.** The output template and search tokens from the last encoder layer are reshaped to the 2D feature maps according to their original spatial positions shown in Fig.2. We input the feature maps to fully convolutional networks (FCN) and concatenate the processed features, generating positional embeddings through linear layers. The search feature map is also taken as the input of to a convolutional head for target bounding box prediction. Finally, we calculate the loss and perform back-propagation to optimize the model parameters.

### 3.2 SELF-SUPERVISED POSITION LOSS

Self-supervised position loss function (SPL) enables each token within a model to implicitly integrate its positional information, thereby enhancing the overall understanding of spatial relationships in the data. As shown in Fig.3, we introduce two distinct types of positional labels: absolute positional labels, which provide the exact location of each token within the input space, and positive-negative positional labels, which indicate whether a token is in a positive or negative position relative to its context.

To implement this, we take the features extracted from both the template and the search regions generated by the tracking model and feed them into two separate fully convolutional networks (FCNs). We denote the template and search feature maps as $E_T \in \mathbb{R}^{C \times H \times W}$ and $E_S \in \mathbb{R}^{C \times H \times W}$. Each FCN is designed with a structure comprising $L$ stacked Conv-BN-ReLU layers, which allows the networks to effectively capture and process the spatial characteristics of the input features. After that, we concatenate the processed template and search features as the positional feature vector:

$$[\mathrm{p}_1; \mathrm{p}_2; \cdots; \mathrm{p}_{N_z+N_x}] = [\mathrm{FCN}(E_T); \mathrm{FCN}(E_S)], \tag{1}$$

Figure 3: The process of self-supervised position loss.

Then, input the feature into absolute and positive-negative positional classifiers:

$$P_{\mathrm{p},y} = \frac{\exp\left(w^\top \mathrm{p}\right)}{\sum_{j=1}^{N} \exp\left(w_j^\top \mathrm{p}\right)}, \tag{2}$$

where $p$ is positional feature; $y$ is positional label; $N$ is the length of the positional feature; $[\mathrm{w}_1; \mathrm{w}_2; \cdots; \mathrm{w}_N] \in \mathbb{R}^{D \times N}$ is the weights in the positional classifier, $D$ denotes the dimension of the feature. These position embeddings not only contain information about the features themselves but also effectively incorporate positional information, enabling the model to better understand and utilize spatial relationships when performing tasks.

Once we have the positional information output from the FCNs, we proceed to compute the position losses using both types of labels. This computation is performed separately for the absolute positional labels and the positive-negative positional labels. By distinguishing the positions of different tokens and their associated positive and negative values, we aim to enhance the semantic consistency between the positions and the features represented by the tokens:

$$L_{abs} = -\frac{1}{n} \sum_{i=1}^{n} \log\left(P_{\mathrm{P}i, y_i^{abs}}^{abs}\right), \tag{3}$$

$$L_{pn} = -\frac{1}{n} \sum_{i=1}^{n} \log\left(P_{\mathrm{P}i, y_i^{pn}}^{pn}\right). \tag{4}$$

The integration of these position losses into the training process serves to reinforce the model's understanding of spatial relationships and improve its ability to accurately track objects across frames. By leveraging self-supervised learning mechanisms, our approach fosters a more robust and nuanced representation of positional information, ultimately leading to better performance in tasks that require precise localization and tracking.

### 3.3 FORCE-DIRECTED ATTENTION

Force-Directed Attention (FDA) innovatively adjusts the distribution of attention among features by introducing the concept of force from physics, enabling the model to more effectively capture meaningful relationships when faced with complex features. FDA defines two interacting forces: repulsive force and attractive force by calculating the differences between queries and keys. Fig.4 illustrates an example of the FDA.

In FDA, the role of the repulsive force is to decrease the attention scores between different features. When the difference between the query and the key is large, the model applies repulsive force, causing these features to be suppressed in the attention distribution. This mechanism effectively reduces the model's focus on irrelevant features, thereby avoiding information interference and enhancing the model's sensitivity to important features. As Eq.5 shown, the repulsive force is calculated by determining the Euclidean distance between the query and the key, squaring it, and then scaling it with the hyperparameter $\delta$ to derive the repulsive force value $A_{repulsive}$. This value plays a suppressive

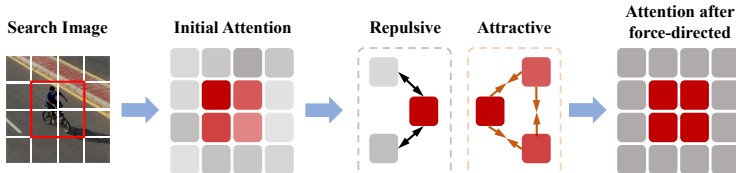

Figure 4: An illustration for Force-directed Attention Mechanism. The red blocks represent the target, while the gray blocks represent the background. Nodes with the same color indicate similar features, which attract each other, whereas nodes with different colors repel one another. Through force-directed modulation, the features of both colors are strengthened during the propagation process.

role in updating the attention scores, resulting in a decrease in the final attention scores $A_{final}$ for features that are significantly different from the query.

$$A_{repulsive} = (\|q_1 - k_1\|_2^2 + \|q_2 - k_2\|_2^2) \cdot \delta, \tag{5}$$

where $q_1$ and $q_2$, as well as $k_1$ and $k_2$ represent different dimensions. We compute the differences across these dimensions separately to capture the feature relationships more comprehensively.

On the other hand, the attractive force enhances the attention scores between similar features. When the difference between the query and the key is small, the model applies attractive force, prompting an increase in the attention scores among similar features. The interaction between targets leads to a situation where the greater the repulsive force, the smaller the attractive force, and vice versa. This process applys an exponential function to the repulsive force, yielding an attractive force value $A_{attractive}$. The introduction of attractive force allows the model to better focus on semantically related features, thereby improving the interrelatedness of features and the efficiency of information transfer:

$$A_{attractive} = e^{-A_{repulsive}}. \tag{6}$$

Finally, by combining repulsive and attractive forces, the force-directed attention $A_{final}$ is obtained:

$$A_{final} = A_{origin} + \alpha \cdot A_{attractive} - \beta \cdot A_{repulsive}, \tag{7}$$

where $\alpha$ and $\beta$ are hyperparameter used to adjust attraction repulsion respectively.

Through the dynamic adjustment of repulsive and attractive forces, FDA can effectively reconstruct the attention matrix, enabling the model to better capture semantic information while considering the interactions between features. This mechanism reduces the problem of semantic misalignment among features, allowing FDA to help the model more accurately understand contextual relationships, thereby enhancing the effectiveness of task completion.

### 3.4 TARGET PREDICTION HEAD

During training, we use weighted focal loss (Law & Deng, 2018) to classify and perform bounding box regression by using L1 loss and generalized IoU loss (Rezatofighi et al., 2019). Combining the two position losses we designed, the overall loss function is:

$$L_{track} = L_{focal} + \lambda_{giou}L_{giou} + \lambda_{L_1}L_1 + \lambda_{abs}L_{abs} + \lambda_{pn}L_{pn}, \tag{8}$$

where $\lambda_{L_1} = 5$, $\lambda_{giou} = 2$, $\lambda_{abs} = 0.01$ and $\lambda_{pn} = 0.1$ are the regularization parameters.

## 4 EXPERIMENTS

### 4.1 IMPLEMENTATION

The experiment is conducted on a server with 8 NVIDIA A100 GPUs. We implement our Enhanced Semantic Alignment Transformer Tracker using Python 3.8 and PyTorch 1.9.

Table 1: Comparison with state-of-the-art trackers on five popular benchmarks. The best two results are shown in **bold font**.

| Method | Year | LaSOT | | | GOT-10K | | | TNL2K | | TrackingNet | | | LaSOT$_{ext}$ | | |
|---|---|---|---|---|---|---|---|---|---|---|---|---|---|---|---|
| | | AUC | P$_{Norm}$ | P | AO | SR$_{0.5}$ | SR$_{0.75}$ | AUC | P | AUC | P$_{Norm}$ | P | AUC | P$_{Norm}$ | P |
| SiamPRN++ (Li et al., 2018) | CVPR19 | 49.6 | 56.9 | 49.1 | - | - | - | 41.3 | 41.2 | 73.3 | 80.0 | 69.4 | 34.0 | 41.6 | 39.6 |
| DiMP (Bhat et al., 2019) | ICCV19 | 56.9 | 65.0 | 56.7 | 61.1 | 71.7 | 49.2 | 44.7 | 43.4 | 74.0 | 80.1 | 68.7 | 39.2 | 47.6 | 45.1 |
| ATOM (Danelljan et al., 2019) | CVPR19 | 51.5 | 57.6 | - | - | - | - | 40.1 | 39.2 | - | - | - | - | - | - |
| SiamR-CNN (Voigtlaender et al., 2019) | CVPR20 | 64.8 | 72.2 | - | 64.9 | 72.8 | 59.7 | - | - | 81.2 | 85.4 | 80.0 | - | - | - |
| Ocean (Zhang & Peng, 2020) | ECCV20 | 56.0 | 65.1 | 56.6 | 61.1 | 72.1 | 47.3 | 38.4 | 37.7 | - | - | - | - | - | - |
| TrDiMP (Wang et al., 2021a) | CVPR21 | 63.9 | - | 61.4 | 67.1 | 77.7 | 58.3 | - | - | 78.4 | 83.3 | 73.1 | - | - | - |
| TransT (Chen et al., 2021a) | CVPR21 | 64.9 | 73.8 | 69.0 | 67.1 | 76.8 | 60.9 | - | - | 81.4 | 86.7 | 80.3 | - | - | - |
| AutoMatch (Zhang et al., 2021) | ICCV21 | 58.3 | - | 59.9 | 65.2 | 76.6 | 54.3 | 47.2 | 43.5 | 76.0 | - | 72.6 | - | - | - |
| STARK (Yan et al., 2021a) | ICCV21 | 67.1 | 77.0 | - | 68.8 | 78.1 | 64.1 | - | - | 82.0 | 86.9 | - | - | - | - |
| TransInMo (Guo et al., 2022) | CVPR22 | 65.7 | 76.0 | 70.7 | - | - | - | 52.0 | 52.7 | - | - | - | - | - | - |
| AiATrack (Gao et al., 2022) | ECCV22 | 69.0 | 79.4 | 73.8 | 69.6 | 80.0 | 63.2 | - | - | 82.7 | 87.8 | 80.4 | - | - | - |
| SwinTrack (Lin et al., 2022) | NeurIPS22 | 71.3 | - | 76.5 | 72.4 | 80.5 | 67.8 | 55.9 | 57.1 | 82.5 | 87.0 | 80.4 | 49.1 | - | 55.6 |
| CiteTracker (Li et al., 2023) | ICCV23 | 69.7 | 78.6 | 75.7 | 74.7 | 84.3 | 73.0 | 57.7 | 59.6 | 84.5 | 89.0 | 84.2 | - | - | - |
| MixFormerV2 (Cui et al., 2024) | NeurIPS23 | 70.6 | 80.8 | 76.2 | - | - | - | 57.4 | 58.4 | 83.4 | 88.1 | 81.6 | 50.6 | - | 56.9 |
| MATTrack (Zhao et al., 2023) | CVPR23 | 67.8 | 77.3 | - | 67.7 | 78.4 | - | 51.3 | - | 81.9 | 86.8 | - | - | - | - |
| OmniTracker (Wang et al., 2023) | CVPR23 | 69.1 | 77.3 | 75.4 | - | - | - | 51.3 | - | 83.4 | 86.7 | 82.3 | - | - | - |
| GRM (Gao et al., 2023) | CVPR23 | 69.9 | 79.3 | 75.8 | 73.4 | 82.9 | 70.4 | - | - | 84.0 | 88.7 | 83.3 | - | - | - |
| SeqTrack (Chen et al., 2023) | CVPR23 | 71.5 | 81.1 | 77.8 | 74.5 | 84.3 | 71.4 | 56.4 | - | 83.9 | 88.8 | 83.6 | 50.5 | 61.6 | 57.5 |
| ARTrack (Wei et al., 2023) | CVPR23 | 72.6 | 81.7 | 79.1 | 75.5 | 84.3 | 74.3 | 59.8 | - | 85.1 | 89.1 | 84.8 | 51.9 | 62.0 | 58.5 |
| DATr (Zhao et al., 2024) | WACV24 | 71.0 | 80.7 | 77.5 | 74.2 | 84.1 | 71.1 | - | - | - | - | - | 51.8 | 62.7 | 59.0 |
| STCFormer (Hu et al., 2024) | AAAI24 | 71.5 | 81.5 | 78.0 | 74.3 | 84.2 | 72.6 | 57.7 | 59.0 | - | - | - | 52.0 | 63.0 | 59.6 |
| AQAT (Xie et al., 2024) | CVPR24 | 72.7 | 82.9 | 80.2 | 76.0 | 85.2 | 74.9 | 59.3 | 62.3 | 84.8 | 89.3 | 84.3 | 52.7 | 64.2 | 60.8 |
| OSTrack (Ye et al., 2022) | ECCV22 | 71.1 | 81.1 | 77.6 | 73.7 | 83.2 | 70.8 | 55.9 | - | 83.9 | 88.5 | 82.0 | 50.5 | 61.3 | 57.6 |
| ESAT-ViTBase | Ours | 71.6 | 81.4 | 78.0 | 74.7 | 84.8 | 71.5 | 58.2 | 59.9 | 84.2 | 88.7 | 83.4 | 51.2 | 62.2 | 58.4 |
| ESAT-HiViT | Ours | 72.8 | 82.9 | 80.4 | 75.9 | 85.2 | 74.9 | 59.5 | 62.3 | 85.2 | 89.6 | 84.7 | 53.5 | 64.6 | 61.5 |

**Model.** Similar to OSTrack, our model consists of 12 sequential encoder layers, initialized with a MAE pre-trained model for the backbone network, where the search area pixels are $384 \times 384$ and the template pixels are $192 \times 192$. In addition, we also conducted experiments using HiViT (Zhang et al., 2023) as the backbone network, which has a total of 20 encoding layers and specific parameter settings consistent with AQAT (Xie et al., 2024).

**Training.** We train our model with following datasets: COCO (Lin et al., 2014), LaSOT (Fan et al., 2018), GOT-10k (Huang et al., 2018) and TrackingNet (Müller et al., 2018). During training process, we set the batch size to 16, the initial learning rate of the backbone to $4 \times 10^{-5}$, the learning rate of other parameters to $4 \times 10^{-4}$, and weight decay to $10^{-4}$. The total number of training epochs is 300, and after 240 epochs, we decrease the learning rate by a factor of 10. More training details can be found in the Appendix A.

**Inference.** During inference, we update the parameters normalization according to loss. The benchmarks we set for test are LaSOT, GOT-10k, TNL2K, TrackingNet and LaSOT$_{ext}$.

## 4.2 RESULTS AND COMPARISONS

To verify the efficacy of the proposed model, we compare them with 23 state-of-the-art approaches on five different benchmarks. Results are shown in Tab. 1.

**LaSOT (Fan et al., 2018).** LaSOT focuses on long-term tracking scenarios, providing a large-scale dataset that includes over 1400 video clips and more than 3.5 million frames, covering 70 categories. Our ESAT-ViTBase performs 0.5% higher than baseline OSTrack-384 in AUC as well as 0.3% higher in P$_{Norm}$ and 0.4% higher in P, which means our method is suitable in long-term video sequences. ESAT-HiViT achieves 72.8% in AUC, 82.9% in P$_{Norm}$ score and 80.4% in P score.

**GOT-10k (Huang et al., 2018).** GOT-10k is a dataset for general object tracking, including over 10000 video clips. It takes the average overlap (AO) and the success rate (SR) at overlap thresholds 0.5 and 0.75 as the evaluation metrics. We obtain improvements of the AO, SR$_{0.5}$, SR$_{0.75}$ of 1.0%, 1.6%, 0.7% compared with OSTrack-384, respectively. ESAT-HiViT obtains 75.9% in AO, 85.2% in SR$_{0.5}$ score and 74.9% in SR$_{0.75}$. We have a significant advantage over most other methods.

**TNL2K (Wang et al., 2021b).** TNL2K is a new dataset for natural language guided tracking, which contains 700 high diversity sequences and introduces several adversarial samples and thermal images to improve the generality of tracking evaluation. Therefore, TNL2K is a challenging benchmark currently. ESAT-ViTBase achieves 58.2% in AUC, which is 2.3% higher than our baseline OSTrack-

Table 2: Quantitative comparison results of tracker with different components. The best results are shown in **bold font**.

| OSTrack-384 | ABS Position Loss | PN Position Loss | FDA | GOT-10K | TNL2K | LaSOT | Speed(fps) | MACs(G) | Params (M) |
|---|---|---|---|---|---|---|---|---|---|
| ✓ | | | | 73.7 | 55.9 | 71.1 | 114.4 | 48.4 | 92.1 |
| ✓ | ✓ | | | 74.0 | 58.0 | 71.3 | 106.6 | 50.9 | 99.0 |
| ✓ | | ✓ | | 74.3 | 57.9 | 71.3 | 106.8 | 50.9 | 99.0 |
| ✓ | | | ✓ | 73.6 | 57.6 | 71.3 | 96.2 | 48.4 | 92.1 |
| ✓ | ✓ | ✓ | | **74.7** | 58.0 | 71.5 | 107.5 | 50.9 | 99.0 |
| ✓ | ✓ | | ✓ | 73.9 | 57.9 | 71.4 | 82.3 | 50.3 | 94.8 |
| ✓ | | ✓ | ✓ | 74.2 | 57.7 | 71.3 | 81.7 | 50.3 | 94.8 |
| ✓ | ✓ | ✓ | ✓ | **74.7** | **58.2** | **71.6** | 83.2 | 50.3 | 94.8 |

384. ESAT-HiViT achieves 59.5% in AUC and 62.3% in P, exceeding almost state-of-the-art tracking algorithms.

**TrackingNet (Muller et al., 2018).** TrackingNet is a large-scale dataset aimed at evaluating the performance of target tracking algorithms in real-world scenarios. It contains over 30000 video clips, covering a variety of different scenes and targets. On TrackingNet, our method outperforms favorably against most other trackers. ESAT-ViTBase achieves 84.2% in success score, 88.7% in $P_{Norm}$ score and 83.4% in P score, while ESAT-HiViT achieves 85.2% in AUC, 89.6% in $P_{Norm}$ score and 84.7% in P score, overtaking most previously published trackers.

**LaSOT$_{ext}$ (Fan et al., 2020).** LaSOT$_{ext}$ is an extended target tracking dataset based on the LaSOT dataset, which includes 150 additional sequences of 15 object classes. Due to its late release, there were relatively few results, but we still have a significant improvement compared to the baseline. For ESAT-ViTBase, our AUC, P and $P_{Norm}$ have increased by 0.7%, 0.8% and 0.9%, respectively. For ESAT-HiViT, we created a new state-of-art on it with an AUC of 53.5%, P of 61.5%, and $P_{Norm}$ of 64.6%.

### 4.3 ABLATION STUDY

In this section, we conducted ablation experiments using ESAT-ViTBase. Effect of values on $\alpha$ and $\beta$, and ESAT-HiViT related ablation experiments are included in Appendix A.

**The Gains of Each Module.** We explored the situations when three modules are used separately or in combination on three benchmarks to judge the exact impact of each part. More specifically, all three parts have a significant improvement on TNL2K and LaSOT whether used alone or in combination. The improvement on GOT-10K is evident when using both types of position loss functions. Tab. 2 shows the results, which prove that each module has made a contribution to the improvement of performance.

**Speed and Size.** Our ESAT runs at a reduced speed compared to the baseline. As shown in the Tab. 2, the parameters of our ESAT are 94.8M, which are 2.7M larger than the original OSTrack-384. As for multiply-accumulate computations (MACs), ours are 50.3G, about 1.9G larger than the baseline. Experiments show that there is not too much computational burden of our method. The addition of FDA will lead to a decrease in running speed and a little increase in the computational burden of parameters and MACs, while the addition for position learning information almost cause no extra burden in speed in comparison to the baseline OSTrack-384.

**Effect of weights on each position loss.** We try different weights for $\lambda$ for each loss we design. The range of testing weights of each loss is approximately determined by the scale of their values. According to Tab. 3, results show that 0.01 and 0.1 are best for absolute position loss and positive-negative position loss, whether used separately or together.

**Visualization.** Fig.5 exhibits some examples of our realtime tracking, and we can see that ESAT is robust in most cases and is able to deal with many challenges. It is able to catch small target (row 3) and distinguish the target from objects that look similar like it (row 1 and row 4). It can also track accurately when there are changes in lighting conditions (row 2) and deal with deformation of the target (row 5).

Table 3: Effect of different weights for ABS position loss and PN position loss. The best results are shown in **bold font**.

| ABS Position Loss | PN Position Loss | GOT-10K | TNL2K | LaSOT |
|---|---|---|---|---|
| 0.001 | - | 73.9 | 57.6 | 71.1 |
| 0.01 | - | **74.0** | 58.0 | **71.3** |
| 0.1 | - | 73.5 | **58.1** | 70.9 |
| - | 0.01 | 73.8 | **57.9** | 71.2 |
| - | 0.1 | 74.3 | **57.9** | 71.3 |
| - | 1 | 73.6 | 57.6 | 71.1 |
| 0.01 | 0.01 | 73.5 | 57.8 | 71.1 |
| 0.01 | 0.1 | **74.7** | **58.0** | **71.5** |
| 0.1 | 0.01 | 73.6 | 57.8 | 70.8 |
| 0.1 | 0.1 | 74.1 | 58.0 | 71.3 |

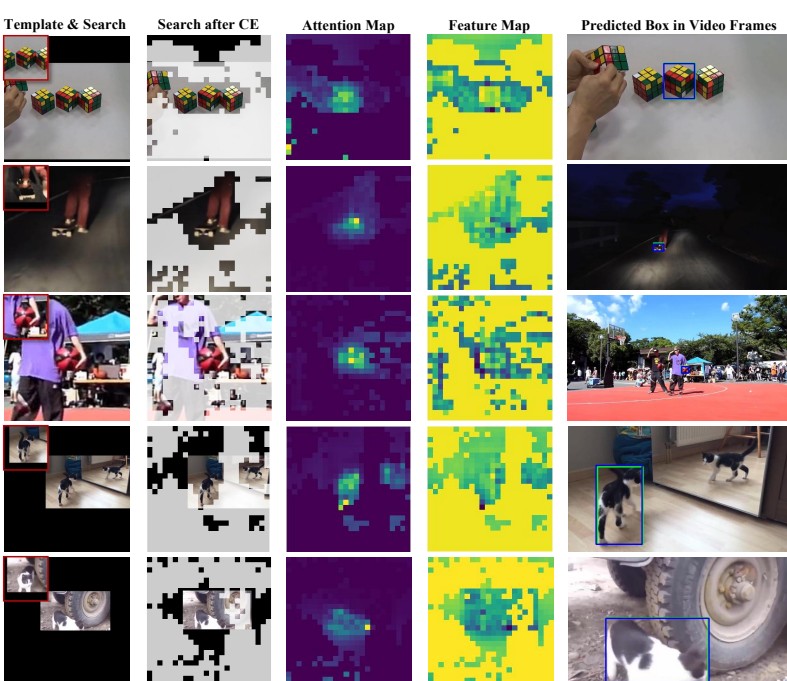

Figure 5: Visualization of the tracking process. The first column is the search images (the big ones) and their template images (small ones on upper left corner). The second column shows the search images after the early candidate elimination process of OSTrack. The third column shows attention map for corresponding search image. The fourth column shows our feature map for this search. The fifth column is the tracking results on corresponding frames. The green rectangles are groundtruth and the blue rectangles are our predicted box.

## 5 CONCLUSION

In this paper, we propose a novel Enhanced Semantic Alignment Transformer Tracker (ESAT) to alleviate the problem of semantic misalignment. Specifically, we design the self-supervised position loss (SPL) and force-directed attention (FDA). SPL learns absolute positional encoding of both target and search regions, and distinguish the target and background through positive-negative labels. FDA uses repulsive and attractive forces to adjust attention and enhance semantic relationships among features. The two modules cooperate to improve semantic alignment. By embedding these modules into ESAT, extensive and quantitative experiments demonstrate the effectiveness of our approach. We hope that our work can provide new insights into addressing challenges related to semantic misalignment in Transformer-based trackers.

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

# A APPENDIX

## A.1 MORE IMPLEMENTATION DETAILS

**Training Details.**

For ESAT-ViTBase, we set the batch size to 16, the initial learning rate of the backbone to $4 \times 10^{-5}$, the learning rate of other parameters to $4 \times 10^{-4}$, and weight decay to $10^{-4}$. The total number of training epochs is 300, and after 240 epochs, we decrease the learning rate by a factor of 10.

For ESAT HiViT, we set the batch size to 4, the initial learning rate of the backbone network to $2 \times 10^{-5}$, , the learning rate of other parameters to $2 \times 10^{-4}$, and the weight decay to $10^{-4}$. The total number of training epochs is 150, and after 120 epochs, we reduced the learning rate by 10 times. The model is trained with AdamW optimizer.

For the GOT-10k (Huang et al., 2018), which requires training the models with only the training split of GOT-10k (one-shot setting), we set the total training epoch to 100 and decrease the learning rate by a factor of 10 after 80 epochs. The other settings are kept consistent with the models trained with all datasets.

## A.2 MORE ABLATION STUDIES

**Effect of values on hyperparameter $\alpha$ and $\beta$.** Due to space limitations in the main text, we will discuss the values of these two hyperparameters here. We try different values on hyperparameter $\alpha$ and $\beta$ for $A_{attractive}$ and $A_{repulsive}$. Tab. 4 shows that the best result is when both $\alpha$ and $\beta$ are set to 0.1 under two types of backbone networks.

Table 4: Effect of values on hyperparameter $\alpha$ and $\beta$. The best results are shown in **bold font**.

| $\alpha$ | $\beta$ | ESAT-ViTBase | | ESAT-HiViT | |
|---|---|---|---|---|---|
| | | TNL2K | LaSOT | TNL2K | LaSOT |
| 0.01 | 0.01 | 58.0 | 71.5 | 59.4 | 72.7 |
| 0.01 | 0.1 | 57.6 | 71.2 | 59.3 | 72.5 |
| 0.1 | 0.01 | 58.1 | 71.4 | 59.2 | 72.3 |
| 0.1 | 0.1 | **58.2** | **71.6** | **59.5** | **72.8** |
| 0.1 | 1 | 57.9 | 71.3 | 59.3 | 72.5 |
| 1 | 0.1 | 57.7 | 71.3 | 59.1 | 72.4 |
| 1 | 1 | 57.8 | 71.4 | 59.2 | 72.4 |

**The contributions of Each Module in ESAT-HiViT.** We also discuss the contribution of each module when the backbone network is HiViT as shown in Tab. 5. Whether used alone or in combination, these three modules have improved tracking performance. We find that FDA effect become more pronounced compared to ESAT-ViTBase, which may be due to the removal of the candidate elimination module when using HiViT as the backbone network. This module may have weakened the effectiveness of FDA.

Table 5: Quantitative comparison results of tracker with different components for ESAT-HiViT. The best results are shown in **bold font**.

| HiViT | ABS Position Loss | PN Position Loss | FDA | GOT-10K | TNL2K | LaSOT | LaSOT$_{ext}$ |
|:---:|:---:|:---:|:---:|:---:|:---:|:---:|:---:|
| ✓ | | | | 74.6 | 58.6 | 71.8 | 52.1 |
| ✓ | ✓ | | | 75.1 | 59.0 | 72.0 | 52.9 |
| ✓ | | ✓ | | 74.9 | 59.1 | 72.2 | 53.1 |
| ✓ | | | ✓ | 75.3 | 59.3 | 72.3 | 52.8 |
| ✓ | ✓ | ✓ | | 75.6 | 59.3 | 72.6 | 53.3 |
| ✓ | ✓ | | ✓ | 75.7 | 59.4 | 72.5 | 53.1 |
| ✓ | | ✓ | ✓ | 75.4 | 59.4 | 72.6 | 53.4 |
| ✓ | ✓ | ✓ | ✓ | **75.9** | **59.5** | **72.8** | **53.5** |

