# OpenReview forum: "Enhanced Semantic Alignment in Transformer Tracking via Position Learning and Force-Directed Attention"
_ICLR.cc/2025/Conference — Submitted to ICLR 2025_

### Official Review · Reviewer_fKCk · 2024-10-22

**Soundness:** 2
**Presentation:** 1
**Contribution:** 2
**Rating:** 5
**Confidence:** 4

**Summary:**

This work focuses on the problem of semantic misalignment in visual object tracking. To address the issues of semantic misalignment caused by position encoding interaction and the inability of traditional attention mechanisms to distinguish semantic attraction and repulsion, the authors propose the Enhanced Semantic Alignment Transformer Tracker (ESAT). This work utilizes self - supervised position loss (SPL) to learn absolute position encodings of the target and search branches and uses force-directed attention (FDA) to adjust the attention weight matrix. For each token, SPL provides two types of positional labels to enhance semantic consistency. FDA defines repulsive and attractive forces to improve feature discrimination. Experiments on multiple public tracking datasets validate the effectiveness of ESAT.

**Strengths:**

1, The authors describe the Enhanced Semantic Alignment Transformer Tracker (ESAT) method, which combines self-supervised position loss (SPL) and force-directed attention (FDA) mechanisms to address semantic misalignment.
2, The paper points out a reasonable theoretical basis. It analyzes the limitations of existing Transformer tracking methods, like semantic misalignment from position encoding and the inability of traditional attention to distinguish semantic attraction and repulsion, and then proposes corresponding solutions.
3, The adequate evaluation conducted numerous experiments on multiple public tracking datasets, demonstrating the effectiveness of the proposed method.

**Weaknesses:**

1、This article lacks the ability to generalize the discussion method on other CNN-Transformer trackers or CNN-based trackers.

**Questions:**

1、 The ESAE layer lacks experimental studies on the number of layers needed to track performance. For the hyperparameters α and β, the author sets the same value. Is a different combination better? There is a misrepresentation: the best result should be 0.1.
2、In addition, although this paper studies the influence of parameters and computation of different modules, it lacks comparison with other representative Transformer-based methods. At the same time, it is noted that after adding the proposed method, the overall reasoning speed is reduced from the original 114.4 fps to 83.2 fps, which decreases by nearly a quarter, and this influence is relatively apparent.
3、This paper proposes a self-supervised position combined with absolute and positive-negative positional encoding. Still, it lacks the comparisons with existing some position encoding ways, such as CPE[1] and so on.
[1] Conditional positional encodings for vision transformers
4、Compared to the SOTA methods, the performance improvement on some datasets, such as LaSOT and LOSOText, is not significant.

---

### Official Review · Reviewer_E6sq · 2024-11-02

**Soundness:** 2
**Presentation:** 2
**Contribution:** 2
**Rating:** 5
**Confidence:** 4

**Summary:**

This paper studies the problems in Transformer-based visual object tracking methods. In particular, the misalignment of the semantic regions and the limitation in attentional modeling have been studied thoroughly. By addressing these issues, the authors have introduced an enhanced semantic alignment learning method and force-directed attention. The experiments show promising results in various datasets.

**Strengths:**

Good writing. Promising results. Comprehensive formulation of the methodology.

**Weaknesses:**

I have many concerns about the presentation, especially the explanation of various terms and concepts. I also have doubts about the novelty and motivation. Please see my questions below.

**Questions:**

(1) My first question is, the 2 studied problems - semantic misalignment and the limitation in attention modeling - do not seem quite related. This makes this study not very concrete, losing the focus on a specific research gap. Furthermore, the proposed force-directed attention does not show too much improvement in Table 2. For example, the FDA does not improve the GOT performance at all. This makes me doubt the necessity of this module.

(2) Regarding the semantic misalignment, the explanation of how the proposed method tackles the raised problem is not clear to me as well. It is confusing to me how the self-supervised position loss can narrow the gap between the target template and each searching frame. I cannot find a comprehensive logical justification for this. I suppose the loss can help the Transformer focus more on foreground areas?

(3) The figure 1 is not very informative. The authors only illustrate several different positional embeddings or features. Even with the caption, I think I cannot understand what refers to the misalignment problem and what refers to the related solution. The figure 2 also does not add too many details on the motivation, only presenting the pipeline structure.

Despite the clean paper structure, I think the research question and motivation of this study are not very clearly defined and discussed. I would like to see authors' explanations on my questions before making my final decision.

---

### Official Review · Reviewer_BZXc · 2024-11-03

**Soundness:** 3
**Presentation:** 2
**Contribution:** 3
**Rating:** 6
**Confidence:** 3

**Summary:**

This paper introduces an Enhanced Semantic Alignment Transformer Tracker (ESAT) to deal with the issue of semantic misalignment by designing a self-supervised position loss (SPL) and a force-directed attention (FDA).

**Strengths:**

The motivation of this paper is clear and it introduces the concept of force from physics.

**Weaknesses:**

The detail of the proposed FDA mechanism is not clear.

**Questions:**

1. The existing related works are not introduced comprehensively, particularly the state-of-the-art models.
2. What is the relationship between Losspos with Lossabs, Losspn in Figures 2 and 3? If Losspos is composed of Lossabs and Losspn, the input of Lossabs and Losspn contains the positional label and the position classifier in Figure 3, but the input of Losspos only contains the position classifier in Figure 2. They are inconsistent.
3. Authors do not describe the detail of FDA mechanism. For instance, what is the initial attention? What are the meanings of q1, q2, k1, and k2 in formula (5)?
4. In Table 4, the values of α and β are always same. Why?

---

### Official Review · Reviewer_iHne · 2024-11-04

**Soundness:** 2
**Presentation:** 3
**Contribution:** 2
**Rating:** 3
**Confidence:** 5

**Summary:**

This paper proposes an Enhanced Semantic Alignment Transformer Tracker, named ESAT. The ESAT use position learning to distinguishing the locations of various tokens and their positive or negative relationships, and force-directed attention to improve feature discrimination. Experimental results on multiple benchmarks show the effectiveness of the proposed ESAT.

**Strengths:**

1.The writing is clear and easy to follow.
2.Ablation experiments show that the method proposed in the paper is relatively effective.

**Weaknesses:**

1.The paper does not compare with some SOTA trackers, such as ARTrack, ARTrackv2, SeqTrack, ODTrack, etc.
2.The performance improvement is limited compared to the baseline OSTrack-384.
3.The position encoding proposed in the paper is an incremental improvement over previous methods, and the performance enhancement is limited. Additionally, the performance improvement of the force-directed attention module is also relatively limited. In summary, the reviewer believes that the originality of this work does not meet expectations.

**Questions:**

1.Is the force-directed attention module proposed in the paper effective?  In Table 2, the comparisons between #5 and #8 do not demonstrate the module's impact.
2.The ABS Position Classifier is designed to learn absolute position encoding. Is there any fundamental difference between this and using learnable position encoding, apart from the distinction between explicit and implicit method?
3.The PN Position Classifier is intended to locate the position of the target within the search area, which should serve a similar purpose as the score map in the prediction head.

**Details Of Ethics Concerns:**

Is the module proposed in the paper really effective. The ablation study shows that the force-directed attention not improve the performance, The performance improvement brought by PN and ABS Position Classifier is limited, and it introduces hyperparameters that need to be adjusted.

---

### Comment · Reviewer_fKCk · 2024-12-02

Summary:
This paper introduces an enhanced semantic alignment method ESAT, which handles the semantic misalignment problem by designing self-supervised position loss (SPL) and force-directed attention (FDA), and obtains excellent results through a large number of experimental results.

Soundness: 3: good

Presentation: 3: good

Contribution: 3: good

Strengths:
The experiments in this paper are very comprehensive and the results are generally good.

Weaknesses:
The introduction and explanation of key components in the manuscript still need to be improved.

Questions:
1. Although the author explained the representation and embedding of features in Figure 2, the description and introduction of the misalignment problem is still not clear and intuitive enough. The same is true for Figure 3. It is recommended to add more details in the picture and related concepts and explanations.
2. The FDA module did not show superior performance in the initial experiments. Although the FDA performance was shown in the supplementary experiments, the explanation of the reasons is still not clear enough.

Flag For Ethics Review: No ethics review needed.

Rating: 6:

Confidence: 5: You are absolutely certain about your assessment. You are very familiar with the related work and checked the math/other details carefully.

Code Of Conduct: Yes

---

> ### Author Response · Authors · 2024-12-03
> **Thank you for your reply!**
>
> Dear Reviewer fKCk,
>
> We are very pleasantly surprised to see your updated comments and the decision to raise the score. Your comments and suggestions are a great encouragement and recognition for us, and they are very helpful in improving the quality of our paper.
>
> However, it seems that your score revision has not been successfully updated in the system. Could you please edit the existing review to update the score? This is very important to us, and we appreciate your help.
>
> Best regards,
>
> Authors

---

### Meta-Review · Area_Chair_QjYL · 2024-12-20

**Metareview:**

The authors have conducted extensive experiments, which yield good overall results. However, the performance improvement observed on certain datasets is not as significant, which may raise concerns about the generalizability of the proposed method. Regarding the methodology, the description and introduction of the misalignment problem lack clarity and intuitiveness. The readers might find it difficult to fully grasp the issue at hand due to the insufficient explanation provided. Furthermore, the force-directed attention module does not appear to function optimally, which could be a critical weakness in the proposed approach. Additionally, the introduction of the ABS Position Classifier does not bring a highly novel concept to the field of tracking, as it seems to be an idea that has been previously explored. A significant concern is the apparent disconnect between the issues of semantic misalignment and the limitations in attention modeling. This lack of a clear relationship between the two makes the study less focused and concrete, potentially obscuring the specific research gap that the paper aims to address.

In summary, while the paper presents a broad range of experiments with promising outcomes, there are issues with the clarity of problem definition, the effectiveness of the proposed modules, and the novelty of the ideas presented. The authors would benefit from refining the focus of their research and providing more compelling evidence of the improvements and innovations claimed.

**Additional Comments On Reviewer Discussion:**

The paper’s extensive experiments yield generally good results, but improvements on some datasets are modest. The explanation of the misalignment problem is unclear, and the force-directed attention module underperforms. The ABS Position Classifier is not particularly innovative. The study’s focus is blurred by an apparent disconnect between semantic misalignment and attention modeling issues, making the research gap less defined. The paper needs a clearer focus and stronger evidence of innovation.

---

### Decision · Program_Chairs · 2025-01-22

Reject